# Systemic Optimization of Gene Electrotransfer Protocol Using Hard-to-Transfect UT-7 Cell Line as a Model

**DOI:** 10.3390/biomedicines10112687

**Published:** 2022-10-24

**Authors:** Roberta Vadeikienė, Baltramiejus Jakštys, Rasa Ugenskienė, Saulius Šatkauskas, Elona Juozaitytė

**Affiliations:** 1Oncology Research Laboratory, Institute of Oncology, Lithuanian University of Health Sciences, LT-50161 Kaunas, Lithuania; 2Research on Delivery of Medicine and Genes Cluster, Faculty of Natural Sciences, Vytautas Magnus University, LT-44001 Kaunas, Lithuania; 3Department of Genetics and Molecular Medicine, Lithuanian University of Health Sciences, LT-50161 Kaunas, Lithuania; 4Institute of Oncology, Lithuanian University of Health Sciences, LT-50161 Kaunas, Lithuania

**Keywords:** electroporation, optimization, plasmid DNA transfer, gene electrotransfer (GET), non-adherent cells, UT-7 cell line

## Abstract

Non-adherent cells are difficult to transfect with chemical-mediated delivery methods. Electroporation is an attractive strategy to transfer the molecules of interest into suspension cells. Care must be taken with the viability of the transfected cells since parameters, which increase cell membrane permeability, subsequently increase transfection efficiency, leading to higher cell death indices. We intended to evaluate the distribution of hard-to-transfect UT-7 cells among different subpopulations: transfected/viable, untransfected/viable, transfected/dead, and untransfected/dead populations, for a better understanding of the relation between gene electrotransfer efficacy and cell death. The following electroporation parameters were tested: pulse strength, duration, plasmid DNA concentration, and ZnSO_4_ as DNase inhibitor. BTX T820 square-wave generator was used, and 48 h after electroporation, cells were observed for viability and fluorescence analysis. Increasing pulse strength correlated directly with an increased ratio of pEGFP-positive cells and inversely with cell viability. The best results, representing 21% pEGFP positive/viable cells, were obtained after EP with 1 HV 1400 V/cm pulse of 250 µs duration using 200 µg/mL plasmid concentration. Results demonstrated that plasmid concentration played the most significant role in pEGFP electrotransfer into UT-7 cells. These results can represent a relevant improvement of gene electrotransfer to obtain genetically modified suspension cells for further downstream experiments.

## 1. Introduction

Cell lines are one of the most valuable tools in experimental biology for research development. In addition, cellular models are beneficial for disease research, especially cancer, testing candidate drugs, personalized therapeutic strategies, and signaling pathways in analysis at gene and protein levels [1] In addition, specific cell lines hold the potential for various applications in biotechnology to produce biopharmaceuticals, develop physiological models for any human tissue, and cutting-edge cell-based technologies such as 3-dimensional printing or microfluidic platforms that can be used successfully in clinical practice [2]. 

The application of cellular models in mentioned research is often related to genetic modifications that allow precise control of specific genes. In most cases, genetic manipulations allow silencing or overexpression of the gene, but latter-day gene-editing tools, i.e., CRISPR/Cas9 and TALEN, are used to induce gene-specific insertion or deletion [3]. The improvement of gene modification protocols is related to establishing efficient transfection. However, some conventional transfection methods, such as lipofection, are unsuitable for some cell lines. In addition, there is a wide variation concerning achieved cell viability and the expression of the gene of interest concerning different cell lines [4]. It is known that non-adherent cell lines are difficult to transfect with standard liposomal or calcium phosphate methods [5]. The common reason for problematic transfection using such methods lies under a weaker attachment of the transfection complex, a mixture of the gene of interest with a mostly liposome-based transfection reagent, to the surface of the cells, leading to reduced uptake of the target DNA [6,7,8]. For such reasoning, there is a need for simple, cost-effective, and efficient technology to manipulate hard-to-transfect suspension cell lines. In this case, electroporation (EP) is the only technique successfully used for the high-efficiency transfer of various molecules into non-adherent cell lines [9,10,11,12,13].

Electroporation can be used for inserting any molecules, e.g., drugs, chemicals, or foreign genetic material (DNA or RNA), in cells and is characterized as the application of brief electric pulses to permeabilize the cell plasma membrane. EP is widely used, and much interest has been given to increasing transfection efficacy by varying pulse parameters and electroporation medium properties. Transfection efficiency and viable cell rates are the main parameters to consider. It is well known that increased transfection efficiency results drop in cell viability [14]. Thus, during the experiments, different electroporation parameters are analyzed, including the configuration of the electric pulse regimen (amplitude, duration, voltage, and the number of pulses), buffer solution, and DNA/RNA or drugs concentration, to achieve high levels of electrotransfer with reduced cell death indices.

UT-7 cell line was established in a patient with acute myeloid leukemia [15]. UT-7 cells have been widely used to study gene regulation, signaling pathways, or anticancer drug sensitivity [16,17,18,19]. First, this cell line is dependent on stimulation by interleukin-3, the granulocyte-macrophage colony-stimulating factor (GM-CSF), or erythropoietin (Epo) for growth and survival, and is a great tool for in vitro establishment of novel sublines of UT-7. For instance, Komatsu and colleagues (1993) declared that the UT-7/Epo subline was established after constant culturing of UT-7 cells with Epo. UT-7/Epo showed an elevated level of heme content and the ratio of benzidine-positive staining compared to UT-7 cells [20]. Subsequently, Komatsu et al. (1997) isolated UT-7/GM cells after long-term culturing with GM-CSF, and these novel cells were capable of differentiation to erythroid and megakaryocytic lineages by treatment with Epo and thrombopoietin, respectively [21]. Therefore, UT-7 cell line versatility for modifications may serve as a model for the experiments related to cells’ response to specific drugs, dosing, or predicting drug combinations for better-individualized therapy of acute myeloid leukemia and other oncohematological diseases, e.g., *BCR/ABL* negative myeloproliferative neoplasms or chronic myeloid leukemia. Several significant studies were carried out with the UT-7 cell line and its subtypes to evaluate their response to chemotherapy drugs, anthracyclines, and antimetabolites. Moreover, next to the cell viability analysis, signaling pathways, e.g., MAPK/ERK, PI3K/AKT, KIT, JAK/STAT, and NGF/TRKA, that contribute to blood cancers, are being investigated at gene and protein levels [22,23,24,25,26,27].

However, UT-7 cells are known for their low efficiency of transfection [28]. Since non-adherent cells are notoriously refractory to most transfection methods and electrotransfer of target genes is becoming the most common strategy to introduce a gene of interest into suspension cells. However, most studies in which the UT-7 cells are modified using electroporation did not aim to improve the method itself, so the electroporation regimen presents great variation. Thus, the protocols for gene electrotransfer (GET) vary from manufacturer to manufacturer, or the researchers must experiment by themselves based on similar data found in articles [28,29,30]. For instance, Weil et al. (2002) conducted gene silencing experiments with UT-7 cells and applied three conditions for the electrotransfer: 300 V and 125 µF; 280 V and 250 µF; 260 V and 960 µF. Cells were resuspended in an Opti-MEM medium, mixed with different concentrations of siRNA, and electroporated. The 280 V and 250 µF conditions were the most efficient, and cell viability following electroporation was the highest. For the best results, under mentioned electroporation conditions, the viability of cells reached almost 85%. Further, results indicated that electroporation using 280 V and 250 µF parameters could successfully enable the penetration of siRNA in these hard-to-transfect cells [28]. On the other hand, Takatoku and his colleagues (1997) aimed to transfect the UT-7 cell line with a plasmid construct and applied electroporation using 250 V and 960 µF conditions [29]. Baiocchi et al. (1997) transfected UT-7 cells using electroporation by adding plasmid DNA and pulsing at 200 V, 950 µF [30]. Hence, considering the practice of other research groups, improved transfection performances can be obtained with selective interventions at critical stages in the electrotransfection process.

Lately, nucleofection, a new electroporation-based method, was developed. It is indicated that nucleofection enables DNA to directly enter the nucleus for the high-efficiency transfection and high viability of a wide variety of cells. According to the manufacturer, nucleofection of the UT-7 cell line results in high levels of transfection and cell viability (96% and 85%, respectively) [31]. However, to the extent of our knowledge, only a few studies have been performed where transfection of UT-7 cells was achieved using a nucleofector [32,33,34]. Thus, the nucleofection device does not allow users to control electrotransfection parameters, and the company that markets the technology does not inform its electric pulse parameters [14,35]. Moreover, costs associated with acquiring nucleofector and nucleofection kits are a relevant reason for research laboratories to look for alternatives to transfect suspension cells. To sum up, there is a need for an optimized and cost-effective protocol that can be widely used in laboratories for UT-7 cell line efficient gene electrotransfer.

Hence, in this work, we describe protocol optimization for the electroporation of the UT-7 cell line. In addition, this study will describe the methodologies we have developed for optimizing EP settings to balance the highest possible transfection efficiency with robust cell viability and growth post-gene electrotransfer. UT-7 cells were electroporated using square-wave pulses, following the systemic screening of pulse strength and duration for best GET efficacy. Furthermore, the impact of DNase inhibitors on GET efficacy was tested by electroporating UT-7 cells in the presence of ZnSO_4_. Finally, we evaluated the influence of varying plasmid DNA concentrations. In addition, the design of our methodology allowed us to assess the distribution of cells among four different subpopulations: transfected/viable, untransfected/viable, transfected/dead, and untransfected/dead, which led us to a better understanding of the relation of GET efficacy and cell death of UT-7 cells.

## 2. Materials and Methods

### 2.1. Cell Line

The UT-7 human cell line was purchased from the German Collection of Microorganisms and Cell Cultures (DSMZ, Braunschweig, Germany). Cell line authentication was performed by DSMZ. The UT-7 cell line was maintained in alpha-MEM (Sigma-Aldrich, Saint Louis, MO, USA) supplemented with 20% fetal bovine serum (Gibco, Gaithersburg, MD, USA), 100 μg/mL streptomycin, 100 units/mL penicillin (Gibco, Gaithersburg, MD, USA), and 2 mM L-glutamine (Gibco, Gaithersburg, MD, USA). Recombinant human GM-CSF (Sigma-Aldrich, Saint Louis, MO, USA) was added to UT-7 cells that require the bioactive protein for survival and proliferation at a final concentration of 5 ng/mL. The cells were cultured in a water-jacketed incubator at 37 °C with 5% CO_2_. UT-7 cells were passaged 1:10 twice a week and always 24 h before the experiment. For some of the experiments, the medium was complemented with 80 µM and 100 µM ZnSO_4_ (Sigma-Aldrich, Saint Louis, MO, USA).

### 2.2. Reporter Construct

The pEGFP (3400 bp) (Takara Bio Inc., Shiga, Japan) encoding a green fluorescent protein was used as an indicator for the assessment of transfection efficiency. The plasmid was extracted from competent *Escherichia coli*, DH5-α strain, selected with ampicillin (Sigma-Aldrich, Saint Louis, MO, USA), and purified with the Plasmid Maxi-Prep Kit (Qiagen, Hilden, Germany). The plasmid was produced at a concentration of 1 μg/μL. Before electroporation, plasmid vectors were loaded at a final concentration of 20 μg/mL (unless otherwise explicitly stated).

### 2.3. Electroporation of UT-7 Cell Line

UT-7 cell line was cultured in T75 flasks (TPP, Trasadingen, Switzerland) and pelleted by gentle centrifugation at 300 RCF for 5 min. Cells were counted using a Neubauer hemocytometer (Weber Scientific, Hamilton, OH, USA) under an optical microscope. An appropriate cell number was suspended in a laboratory-made electroporation medium at a concentration of 1.8 × 10^6^ cells/mL. Next, a volume of 50 µL (9 × 10^4^ cells) was transferred between stainless steel plate electrodes separated with a 2 mm gap. A BTX T820 electroporator (Harvard Apparatus) (Artisan Technology Group, Champaign, IL, USA) or laboratory-made electroporator was used for pulsing, changing the voltage, and pulse duration. A square wave 1 HV (i.e., high voltage) pulse, 1200–1800 V/cm pulse strength, and 100–750 µs pulse duration were delivered. After pulsing, UT-7 cells were transferred into 24 well plates (TPP, Trasadingen, Switzerland) and left for 20 min for recovery. After the recovery period, cells were seeded into two 24-well and 96-well plates (TPP, Trasadingen, Switzerland) for transfection, propidium iodide staining, and MTS assays, respectively. 24-well plates were supplemented with a culture medium up to 450 µL and 96-well plates up to 150 µL. Afterwards, plates were placed in an incubator for 48 h. The electroporation regimen was carried out at room temperature conditions.

The characteristics of the laboratory-made EP medium were 0.1 S/m conductivity, 270 mOsmol osmotic pressure, and 7.2 pH. The medium contained 1.73 mM MgCl_2_ (Sigma-Aldrich, Saint Louis, MO, USA), 5.59 mM Na_2_HPO_4_ (Sigma-Aldrich, Saint Louis, MO, USA), 3.00 mM NaH_2_PO_4_ (Sigma-Aldrich, Saint Louis, MO, USA), and 242.20 mM sucrose (Sigma-Aldrich, Saint Louis, MO, USA). EP medium was filtered out through 0.22 µm syringe filters before usage.

### 2.4. Flow Cytometry

UT-7 cell line was observed for viability and fluorescence 48 h post-transfection. The analysis was performed using BD Accuri C6 (BD Biosciences, Franklin Lakes, NJ, USA) flow cytometer. 22 μm core size and 66 μL/min flow rate were used for all measurements.

The cell culture was pelleted by centrifugation at 300 RCF for 3 min in a centrifuge to remove the growth medium. Then, UT-7 cells were resuspended in 200 μL 1xPBS (Sigma-Aldrich, Saint Louis, MO, USA) buffer. For pEGFP transfection efficiency assessment, cells were excited using a 488 nm laser, and data was collected by counting 10,000 events/samples using a 533/30 nm FL1-A emission filter for green color.

Propidium iodide (PI) staining was used to determine dead cells after electroporation. First, the cell samples were centrifugated at 300 RCF for 3 min to remove the growth medium. Afterwards, cells were diluted in 90 µL 1xPBS (Sigma-Aldrich, Saint Louis, MO, USA) containing 10 µL PI (400 µM) solution (Sigma-Aldrich, Saint Louis, MO, USA). 3 min after dilution, UT-7 cells were measured by flow cytometry. Cell permeability to propidium iodide was analyzed using a 585/40 nm FL2-A emission filter for red color. Cells negative for PI were considered living cells.

Color compensation was used by subtracting 6% of the pGFP channel from the PI channel to avoid color spillover of transfected cells into the population of dead cells. Separate controls of only pGFP^+^ and PI^+^ cell populations were acquired for compensation. The gating strategy is depicted in the Results section.

The transfection efficacy was calculated as the ratio of transfected/viable cells vs. the number of viable cells in the control sample (Equation (1)):(1)$$GET=\frac{N_{Sample\;}\times 100\%}{N_{Control}}$$
where *GET* stands for transfection efficacy and *N* for cell number.

### 2.5. MTS Assay

MTS tetrazolium (Cell Titer96 Aqueous; Promega, Madison, WI, USA) assay was performed for metabolism assessment of the UT-7 cell line. After electroporation, UT-7 cells were seeded into a 96-well plate at a density of 9000 cells/well for up to 48 h following previously described conditions. The relative viability of the UT-7 cell line was analyzed by measuring the amount of MTS converted by the cells. Briefly, 40 µL MTS (5mg/mL in 1xPBS) was added to each well and incubated for another 3 h at 37 °C, 5% CO_2_. The absorbance was measured at 490 nm by a microplate reader (Tecan GENios Pro, Grödig, Austria).

### 2.6. Statistical Analysis

IBM SPSS Statistics 22.0 (IBM Corp., Armonk, NY, USA) software was used to perform statistical tests. The data were analyzed using an independent sample t-test and one-way analysis of variance (ANOVA), followed by post hoc Tukey multiple comparison tests when appropriate (Appendix A). The data are presented as means ± standard deviation (SD), while the *p* < 0.05 value was considered statistically significant. Experiments were performed in duplicates and repeated at least two times independently.

## 3. Results

### 3.1. Optimization of the Gene Electrotransfer Conditions

We established a gating strategy for a proper assessment of transfected cells co-stained with PI for distinguishing among four different cell subpopulations in the sample (Figure 1). Using such a strategy, we were able to differentiate among transfected/viable (pEGFP^+^/PI^−^), untransfected/viable (pEGFP^−^/PI^−^), transfected/dead (pEGFP^+^/PI^+^), and untransfected/dead (pEGFP^−^/PI^+^) cell subpopulations.

The gating strategy starts with excluding cells from debris by plotting singlet cells according to their optical properties as forward scattering-area (FSC-A) vs. forward scattering-height (FSC-H) (Figure 1A). Then GFP^+^ (transfected) from untransfected cells (GFP^−^) (Figure 1B) and PI^−^ (viable) cells from PI^+^ (dead) (Figure 1C) cells were separated. We used 6% compensation (Figure 1F) in the GFP channel to avoid the spillover of GFP fluorescence into the PI channel (Figure 1E).

The effect of the EP buffer on transfection efficiency was verified by performing GET in three different electroporation buffers, laboratory-made EP medium, BTXpress Cytoporation Medium T (BTX, Holliston, MA, USA), and Opti-MEM (Gibco, Gaithersburg, MD, USA). A laboratory-made EP medium resulted in the same number of viable and transfected cells compared to different manufacturers’ EP buffers (results not shown). Therefore, the following experiments were performed in a laboratory-made EP medium.

Different electroporation conditions were tested, including variations in electric field strength (1.2, 1.4, 1.6, 1.8 kV/cm) and pulse duration (100, 250, 500, 750 μs). In each experiment, a single high voltage pulse was applied, and 20 μg/mL of the plasmid was used.

We exploited three different assays to determine the viability of UT-7 cells to discriminate between live and dead cells since it is impossible to wash out dead suspension cells from the samples without losing viable ones. For this reason, we used Propidium iodide (PI) to stain dead cells with a leaky plasma membrane. Then, the overall cell number was obtained by counting all cells in the sample using flow cytometry, and PI-positive (dead) cells were excluded from the population. Furthermore, different gene electrotransfer parameters were tested, and the overall cell number, including PI results, was compared to the MTS assay (Figure 2).

Results demonstrated that the cell death rate increased by increasing the voltage and duration of the electric pulse. Moreover, we determined that the overall cell number compared to the number of viable (PI negative) cells was significantly higher throughout the whole range of parameter sets from 1.6 kV/cm 100 µs. Only the 1.2 kV/cm 250 µs parameter set showed no difference (Figure 2). Notably, the difference between overall cell number and amount of PI negative (viable) cells kept increasing significantly from 1.6 kV/cm 500 µs. MTS results which represent cell metabolic activity, were significantly higher only after gene electrotransfer under 1.4 kV/cm 250 µs to 1.4 kV/cm 500 µs excluding 1.2 kV/cm 500 µs parameters. Herewith, cell viability determined using the MTS assay was significantly higher only using 1.4 kV/cm 500 µs, compared to the PI negative (viable) cell amount (65.63 ± 9.35% vs. 40.55 ± 12.99%, *p <* 0.007).

These results confirmed that the best solution for further experiments was to follow the PI staining of dead cells since PI staining and metabolic activity demonstrated similar results. However, the PI method was considered superior to the MTS method in our case due to the ability to use the PI methodology for simultaneous cell viability and transfection efficiency measurements.

Next, we evaluated pEGFP plasmid electrotransfection efficiency in UT-7 cells under the same EP parameter sets (Figure 3) as in Figure 2. Results revealed that around 11% of cells were successfully transfected and were viable at conditions of 1 HV pulse in the range from 1.2 kV/cm to 1.6 kV/cm with 500 µs pulse duration (Figure 3A). Together, these parameters provided one of the highest fluorescence intensities of transfected cells (Figure 3B).

Our data indicate that increasing pulse intensity and duration was the significant cause of the growing number of transfected/dead (pEGFP+/Dead) cells (Figure 3A). Such results suggest that some cells were successfully transfected and could produce enough EGFP protein to be detected but perished in time. Moreover, it was determined that increasing pulse duration correlated directly with an increased ratio of transfected/viable (pEGFP+/Viable) cells (Figure 3A). In contrast, pulse intensity played a less significant role (Figure 3A, comparing different pulse intensities with the same pulse duration vs. the same pulse intensity with varying pulse durations). However, the increase in transfected/viable cell numbers was negligible compared to the increase in cell death rate. The same patterns were observed in transfected/dead and untransfected/dead UT-7 cell subpopulations, where the number of cells in these subpopulations increased with pulse duration rather than pulse intensity.

An opposite pattern was noticed in the untransfected/viable subpopulation, where the number of cells diminished notably with increasing pulse intensity and duration. The same pattern was observed in the overall cell number decrease (Figure 3A). For instance, the overall cell number and number of untransfected/viable cells tended to decrease in a stair-like pattern under the last pulse intensity within the same pulse duration set. For instance, comparing the whole 250 µs parameters set and 1.6 kV/cm 250 µs vs. 1.8 kV/cm 250 µs. A similar pattern repeats with other pulse duration sets, where increasing pulse intensity has low or no impact on the overall cell number, but the last parameter shows a prominent decrease. This data emphasizes the choice complexity of EP parameters for prominent GET into suspension cells such as UT-7.

In contrast, the increase in mean fluorescence intensity in the transfected/viable cell subpopulation was more pronounced than the transfected/viable cell number. These results suggest that increasing pulse strength and duration helps more plasmid molecules enter electroporated cells.

### 3.2. Effect of DNase Inhibitor ZnSO_4_ on the Electroporation Process

Our next step was to evaluate the impact of ZnSO_4_ on GET efficiency under conditions from previous experiments that demonstrated the highest efficiency. Subsequently, UT-7 cells were electroporated with 1.4 kV/cm 250 µs, 1.6 kV/cm 250 µs, 1.8 kV/cm 250 µs, 1.2 kV/cm 500 µs, 1.4 kV/cm 500 µs, and 1.2 kV/cm 750 µs pulses in the presence of 80 µM and 100 µM ZnSO_4_ (Figure 4). The treatment with ZnSO_4_ was provided immediately after GET by pouring ZnSO_4_ on the cells and incubating until data acquisition after 48 h.

ZnSO_4_ acts as an inhibitor of intracellular nucleases and potentially can prevent plasmid degradation after plasmid DNA entry into the cell. However, we determined no supplementary effect of ZnSO_4_ on pEGFP GET efficiency in suspension UT-7 cells. For instance, the overall cell number decreased according to pulse intensity and duration. Here, the subpopulation number in transfected/viable cells remained the same under different EP parameters and varying ZnSO_4_ concentrations (Figure 4A). No significant changes in the mean fluorescence intensity were observed when comparing two different ZnSO_4_ concentrations.

### 3.3. The Concentration of the Plasmid Impact on GET Efficiency

Furthermore, we aimed to evaluate the influence of pEGFP concentration in an electroporation medium on the GET efficiency in UT-7 cells (Figure 5). Various plasmid concentrations were tested, i.e., 20, 50, 100, and 200 µg/mL. Two electric field conditions were used, 1.2 kV/cm and 1.4 kV/cm. A single high voltage pulse of 250 µs duration was applied for the experiment.

Here, we determined that the concentration of pEGFP plasmid is a key factor in increasing GET efficiency in UT-7 cells. The increase of plasmid concentration helped to elevate the amount of transfected/viable (pEGFP+/Viable) cells per sample: 1.2 kV/cm 250 µs 20 µg/mL pEGFP vs. 1.2 kV/cm 250 µs 50 µg/mL pEGFP (7.48 ± 0.71% vs. 8.75 ± 0.79%, *p* < 0.001), 1.2 kV/cm 250 µs 100 µg/mL pEGFP (14.04 ± 1.32%, *p* < 0.001), and 1.2 kV/cm 250 µs 200 µg/mL pEGFP (19.46 ± 3.28%, *p* < 0.001). Where 1.4 kV/cm 250 µs 20 µg/mL pEGFP vs. 1.4 kV/cm 250 µs 100 µg/mL pEGFP gave 6.81 ± 0.58% vs. 14.94 ± 1.68%, *p* < 0.001 and 1.4 kV/cm 250 µs 20 µg/mL pEGFP vs. 1.4 kV/cm 250 µs 200 µg/mL pEGFP (6.81 ± 0.58 vs. 21.07 ± 2.45%, *p* < 0.001).

Further, the results revealed that an increase of pEGFP plasmid from 20 µM to 200 µM had only a minor impact on cell viability, thus sustaining the usage of higher plasmid concentrations in an attempt to obtain a higher number of transfected/viable UT-7 cells.

## 4. Discussion

It is known that electroporation can be an efficient method to introduce plasmid DNA into cells of interest, including those that are often considered as difficult to transfect, e.g., non-adherent cell lines. In this work, we have characterized the optimization of gene electrotransfer protocol for the suspension UT-7 cell line using pEGFP plasmid DNA. It was found that the transfected cell rates and cell viability of UT-7 cells rely on the selected square-wave pulse duration and critically rely on the used plasmid DNA concentration (Figure 6).

Here, we achieved gene electrotransfection rates of 10.59 ± 2.86% and 11.03 ± 3.24%, representing successfully transfected/viable cells (GFP+/PI−) which were obtained when UT-7 cells were electroporated using 1.6 kV/cm 500 µs and 1.2 kV/cm 500 µs pulses in the presence of 20 µg/mL pEGFP plasmid, respectively. To our knowledge, only several studies have demonstrated gene electrotransfer using the UT-7 cell line [28,29,30]. Unfortunately, the protocols described by Weil et al. (2002), Baiocchi et al. (1997), and Takatoku et al. (1997) did not aim for the improvement of the method itself, and electroporation was used just as an application for genetic cell modification. The main difference in our protocol is the variety of electroporation conditions tested. In previously mentioned studies: an electric field of 280 V and a capacitance of 250 µF, 200 V with a capacitance of 950 µF, and 250 V with a capacitance of 960 µF were used, respectively. Thus, the transfection efficiency and cell viability obtained in our research are hard to compare to other authors’ results due to different study designs. Lately, a new electrotransfer method known as nucleofection has been developed and applied for UT-7 cell transfection [15,32,33,34]. It was stated that nucleofection of UT-7 cells results in high levels of transfection and cell viability (96% and 85%, respectively) [31]. However, an association between our protocol and nucleofection is impossible since the conditions of the electric pulse and pulsing buffer composition are not described by the authors or the company that markets the nucleofector.

Having defined the best conditions, we further tested the effect of ZnSO_4_ as a nuclease inhibitor on the electroporation process. Nucleases are responsible for plasmid degradation before entering the nucleus and subsequently lower cell proliferation rates [36,37]. Here, we hypothesized that the efficiency of gene electrotransfer could be improved by using the DNase inhibitor Zn^2+^. However, we did not use other substances, i.e., EDTA or EGTA, as DNase inhibitors because they cause elevated cell death indices and lower transfection efficiency [14]. Thus, EDTA and EGTA are highly conductive mediums, but electroporation is usually performed in a low-conductance medium to avoid the thermal effect that reduces cell viability. To prevent the increase in EP medium conductivity and the impact of Zn^2+^ ions on electropermeabilization and the plasmid DNA–plasma membrane interaction, the ZnSO_4_ was instantly added into the growing medium after the electric pulses were delivered. However, we did not observe any effect of ZnSO_4_ on transfection efficiency and UT-7 cell viability. As we know, there is no report on UT-7 cell line transfection improvement with DNase inhibitors. Only Delgado-Canedo et al. (2006) reported a comprehensive analysis of electrotransfection conditions improvement in the suspension K562 cell line. In the study, better efficiency of electroporation was achieved by using DNase inhibitor ZnSO_4_ at a concentration of 80 µM, adding the reagent immediately after the pulses [14].

Our data suggest that another critical factor for effective gene electrotransfer is the concentration of plasmid DNA added to the cell suspension before pulsing. We evaluated the impact of various amounts of pEGFP, i.e., 20, 50, 100, and 200 µg/mL, on gene electrotransfer efficiency. It was noticed that increased concentration of the plasmid DNA contributed to a drastically elevated amount of transfected/viable UT-7 cells (pEGFP+/Viable) per given sample. Using a 1.4 kV/cm 250 µs pulse with 200 µg/mL pEGFP resulted in the best combination of the cell transfection and viability (where GFP+/Viable cell subpopulation reached 21.07 ± 2.45%, *p* < 0.001). Several studies have reported that an increased plasmid concentration (from 50 to 300 µg/mL) enhances transfection efficiency; however, at the same time, cell viability was reduced significantly [38,39,40]. Notably, these findings were made with non-human, both adherent and non-adherent, cell lines. Based on our optimized protocol, it can be seen that human suspension cell lines respond quite differently to the increase in plasmid concentration used for the GET resulting in more significant amounts of transfected/viable cells without a prominent cell viability decrease.

Therefore, transfection efficacy in this work was represented as a ratio of transfected/viable cells vs. the overall viable cells in the control sample rather than the percentage of transfected cells per sample (Equation (1)). Such a formula will allow the reader to predict in one step how many transfected/viable cells one may expect after the GET procedure following our protocol.

It can be concluded that an increase in electric pulse duration, but not the intensity, elevates gene electrotransfection efficiency into suspension UT-7 cells. However, an increase in pulse intensity and duration substantially diminishes cell viability without a notable increase in transfected/viable cell number. On the other hand, an increase in plasmid concentration considerably enhances GET efficiency with a slight impact on cell viability.

## 5. Conclusions

An increase in HV pulse duration results in more efficient gene electrotransfer into suspension UT-7 cells compared to pulse intensity;Additional tools or conditions should be taken into consideration for achieving higher pDNA electrotransfer into suspension UT-7 cells since increasing HV pulse intensity and duration results in higher cell killing rather than transfected/viable cell number;For instance, an increase in pDNA concentration results in a more prominent increase in transfected/viable cell subpopulation compared to pulse duration and intensity;However, DNAase inhibitor ZnSO4 had no impact on pDNA electrotransfer into suspension UT-7 cells nor affected cell viability.

## Figures and Tables

**Figure 1 biomedicines-10-02687-f001:**
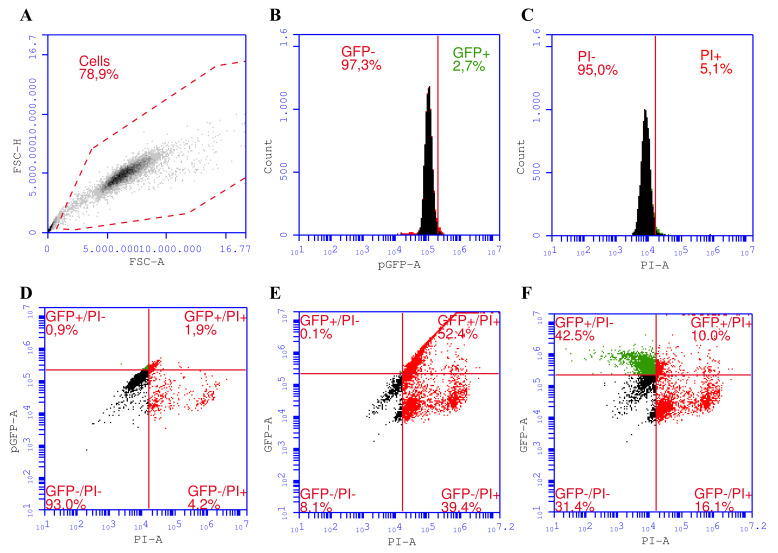
Schematic representation of gating of the samples to distinguish four subpopulations of UT-7 cells representing pEGFP transfected/viable (GFP+/PI−), transfected dead (GFP+/PI+), untransfected/viable (GFP−/PI−) and untransfected/dead (GFP−/PI+) cells. Gating strategy: cells in the control sample according to forward scattering–area (FSC-A) vs. forward scattering–height (FSC-H) (**A**), control sample in GFP channel after 6% compensation (**B**), a control sample in PI channel after 6% compensation (**C**), GFP vs. PI gating of the control sample (**D**), GFP vs. PI gating of the GET sample without compensation (**E**), GFP vs. PI gating of the GET sample after 6% compensation (**F**).

**Figure 2 biomedicines-10-02687-f002:**
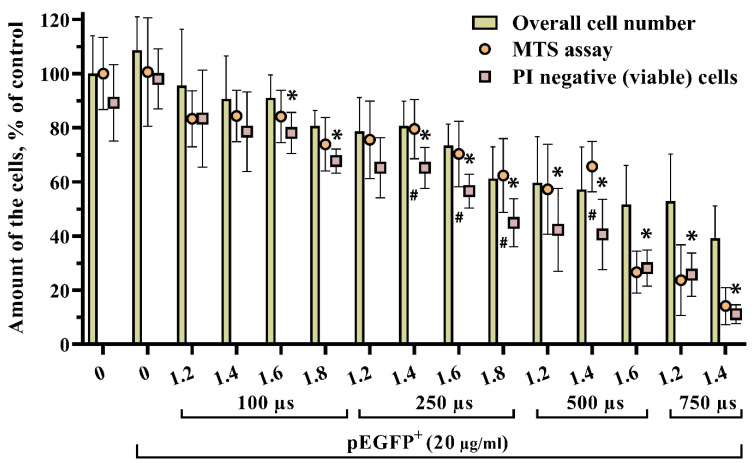
Viability of UT-7 cells after pEGFP gene electrotransfer using different pulse strengths and durations, evaluated by flow cytometry counting, MTS assay, and PI staining. Overall cell number represents the total number of suspension cells per sample obtained by flow cytometry. MTS stands for metabolically active cells expressed as percentage shift in metabolic activity compared to the control samples and cells with normal integrity nonpermeable plasma membrane depicted as PI negative or viable cells. The X-axis shows pulse strength in kV/cm. Error bars represent ± SD from two independent experiments performed in quadruplicate and were further analyzed using one-way ANOVA with Tukey’s multiple comparisons test. Significantly different treatments at *p* < 0.05 are represented as * and #. The * stands for the significant difference between overall cell number and overall viable cells, while # represents a significant difference between cell metabolic activity (MTS) and overall viable cells.

**Figure 3 biomedicines-10-02687-f003:**
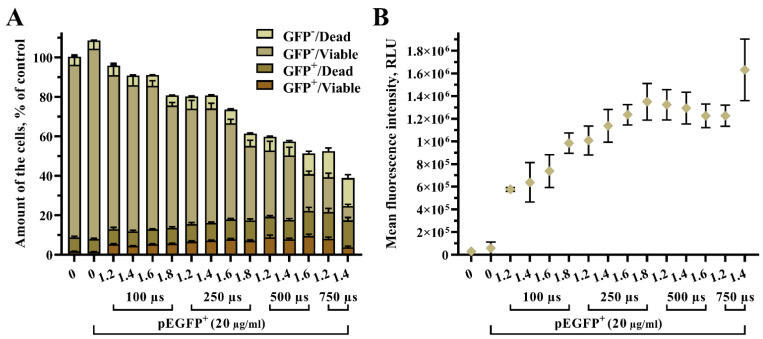
Distribution of UT-7 cells per sample among four groups representing transfected/viable cells denoted as (GFP^+^/Viable), transfected/dead as (GFP^+^/Dead), untransfected/Viable as (GFP^−^/Viable), and untransfected/dead as (GFP^−^/Dead) shown in stacked bars. The overall cell number in the sample is shown as column height (**A**). The pEGFP fluorescence intensity is represented by a scatter dot plot (**B**). The X-axis depicts pulse strength in kV/cm. The data points are shown as mean ± SD from two independent experiments performed in quadruplicate and were further analyzed using one-way ANOVA with Tukey’s multiple comparisons test. All statistical results are presented in the Supplementary file (Appendix A).

**Figure 4 biomedicines-10-02687-f004:**
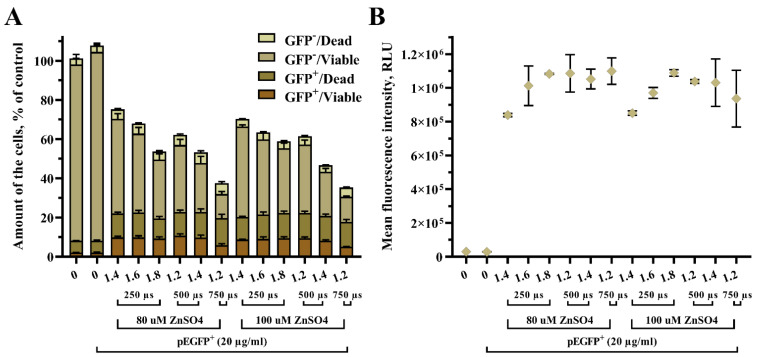
The effect of ZnSO_4_ for pEGFP GET efficiency in UT-7 cells under varying pulse strengths (kV/cm- X-axis) and pulse duration (µs), where transfected/viable cells denoted as (GFP^+^/Viable), transfected/dead as (GFP^+^/Dead), untransfected/viable as (GFP^−^/Viable), and untransfected/dead as (GFP^−^/Dead) shown in stacked bars. The overall cell number in the sample is shown as column height (**A**). The pEGFP fluorescence intensity is represented by a scatter dot plot (**B**). The data points are shown as mean ± SD from two independent experiments performed in quadruplicate and were further analyzed using one-way ANOVA with Tukey’s multiple comparisons test. All statistical results are presented in the Supplementary file (Appendix A).

**Figure 5 biomedicines-10-02687-f005:**
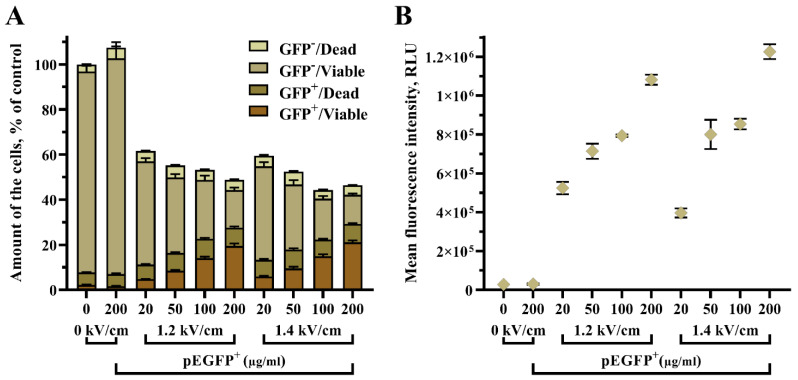
The impact of pEGFP plasmid concentration for the GET efficiency in UT-7 cells, where transfected/viable cells are denoted as (GFP^+^/Viable), transfected/dead as (GFP^+^/Dead), untransfected/viable as (GFP^−^/Viable), and untransfected/dead as (GFP^−^/Dead) shown in stacked bars. The overall cell number in the sample is shown as column height (**A**). The pEGFP fluorescence intensity is represented by a scatter dot plot (**B**). The X-axis depicts pulse strength in kV/cm. The data points are shown as mean ± SD from two independent experiments performed in quadruplicate and were further analyzed using one-way ANOVA with Tukey’s multiple comparisons test. All statistical results are presented in the Supplementary file (Appendix A).

**Figure 6 biomedicines-10-02687-f006:**
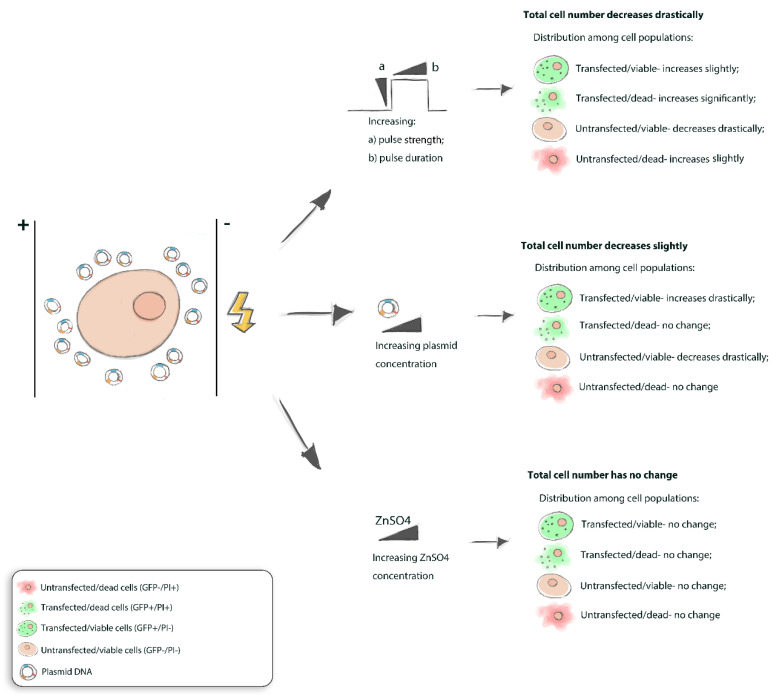
Overall schematic representation underlying the influence of experimental conditions affecting GET efficiency into suspension UT-7 cell line based on the results described. In the figure (**a**) stands for pulse strength, while (**b**) represents pulse duration.

## Data Availability

The data presented in this study are available on request from the corresponding author.

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
