# Peer review of "Systemic Optimization of Gene Electrotransfer Protocol Using Hard-to-Transfect UT-7 Cell Line as a Model"

_biomedicines, 2022, doi:10.3390/biomedicines10112687_

Round 1

Reviewer 2 Report

Nice paper. It should be published

Author Response

Thank you for your patience, attentive reading, and positive feedback. 

Reviewer 3 Report

The article of Roberta Vadeikiene et al, entitled of “Systemic optimization of gene electrotransfer protocol using hard-to-transfect UT-7 cell line as a model”, present a protocol optimization for the electroporation of the UT-7 cell line, developed for optimizing EP settings in order to balance the highest possible transfection efficiency with robust cell viability and growth. It is a well-structured article that describe an interesting study that is helpful for those who are performing transfection assays using suspension cells. However, I have a few minor concerns that should be addressed before the present article is being published:

I believe that the authors should rewrite the “Conclusions” section, in order to highlight the importance of their study before mentioning what else could be tested in the framework of the study.

Figure 6: The particular image is not sharp. Their resolution must be improved.

English language correction is needed. The main text contains some grammatical and syntax errors that need to be corrected.

Author Response

Thank you for your patience, attentive reading, and thoughtful remarks. 

Point 1: I believe that the authors should rewrite the “Conclusions” section, in order to highlight the importance of their study before mentioning what else could be tested in the framework of the study.

Response 1: The conclusions were rewritten with numbers by highlighting each of the obtained results. 

Point 2: Figure 6: The particular image is not sharp. Their resolution must be improved.

Response 2: All figures were reloaded with better resolution. 

Point 3: English language correction is needed. The main text contains some grammatical and syntax errors that need to be corrected.

Response 3: the English language was checked and errors were corrected.